# VistaGUI: Towards More Robust and Intelligent GUI Automation

## Abstract

The proliferation of Large Language Models (LLMs) and Vision-Language Models (VLMs) has driven the development of general-purpose agents for Graphical User Interface (GUI) automation. Despite this progress, the practical application of these agents is hindered by their fragility, which stems from three primary limitations: low retrieval accuracy in retrieval-augmented generation (RAG), over-reliance on single-modality perception, and inadequate failure recovery mechanisms. To address these challenges, we introduce **VistaGUI**, a robust, multi-modal GUI agent that integrates optimized retrieval, adaptive sensing, and environment-aware state management into a unified framework. The core contributions of VistaGUI are threefold. First, a parallel instruction-understanding module enhances retrieval accuracy to better comprehend user intent, enabling more precise, context-aware decision-making. Second, an adaptive multi-modal sensing module dynamically selects the optimal perception modality—including API-based queries, visual perception, and OCR—to achieve a comprehensive understanding of diverse GUI environments. Third, an environment-aware state management system records and analyzes interaction trajectories to proactively detect and efficiently recover from execution failures, thereby reducing replanning overhead. VistaGUI is implemented within a modular architecture comprising a Knowledge Manager, Planner and Action Executor. Extensive experiments conducted on a diverse set of GUI automation tasks demonstrate that VistaGUI significantly outperforms strong baselines in task success rate, recovery speed, and overall robustness.

## 1 Introduction

The rapid advancement of large language models (LLMs) and vision-language models (VLMs) (Li et al., 2023; Deepmind, 2025a;b; OpenAI, 2024a; 2025; Qin et al., 2025b; Xie et al., 2025; Yang et al., 2025) has catalyzed remarkable progress in automating tasks within graphical user interfaces (GUIs), opening new frontiers in human-computer interaction. GUI agents represent a transformative paradigm: they enable natural language-driven automation for activities ranging from simple form filling and system configuration to complex multi-application workflows.

Yet, despite their promise, deploying GUI agents in real-world environments remains deeply challenging. Unlike structured APIs or static datasets, GUIs are inherently *dynamic, diverse, and multimodal*. Interface designs differ dramatically across platforms, layouts change frequently with software updates, and visual elements vary widely in both form and semantics. Moreover, effective automation requires reasoning over heterogeneous inputs: perceptual signals (e.g., screenshots, OCR results) must be integrated with symbolic context (e.g., API responses, action histories). These challenges make GUI automation Xie et al. (2024); Rawles et al. (2025); Bonatti et al. (2024) far more complex than conventional embodied or web-based tasks.

While recent advances have improved GUI agents, they remain far from practical deployment due to a central limitation: the inability to dynamically construct rich and actionable context for decision-making. This shortcoming manifests in three fundamental ways. **(1) Inaccurate Knowledge Retrieval.** To handle diverse tasks, agents (Agashe et al., 2025a) rely on external retrieval engines to interpret ambiguous instructions. Yet retrievals often misrepresent user intent, returning partial or misleading context. These flawed inputs misguide planning and lead to premature failures or

completely erroneous execution paths. **(2) Brittle Environment Perception.** Current perception pipelines are fragmented. Vision-based models (Qin et al., 2025a; Lu et al., 2024) are fragile to minor cosmetic changes, while structure-based methods (Song et al., 2025; Gur et al., 2024) (e.g., DOM queries (He et al., 2024) ) miss layout cues and often cannot access essential elements due to design or privacy restrictions. Accordingly, agents perceive GUIs in a distorted and incomplete way, undermining reliable reasoning. **(3) Lack of Adaptive Error Recovery.** Most agents adopt a naïve "replan-from-failure" strategy (Agashe et al., 2025a;b; Zhang et al., 2025), discarding entire plans in response to small errors. This is akin to throwing away a recipe over one missing ingredient. When a required window (e.g., Settings) disappears due to a glitch, the agent cannot compare the failed state with a prior successful one, nor reuse the simple corrective action. Instead, it is forced into costly replanning from an unstable state, wasting valid knowledge and compounding errors.

**Contribution.** We present **VistaGUI**, a modular multimodal agent that tackles three long-standing challenges in GUI automation—noisy retrieval, brittle perception, and fragile recovery—through a proactive loop of *planning, acting, and reflecting*. VistaGUI introduces novel mechanisms in each phase to align perception, reasoning, and recovery with the realities of dynamic GUI environments.

To address the critical challenge of inaccurate knowledge retrieval, VistaGUI first introduces strategic planning for robust retrieval. Instead of relying on a single, potentially flawed retrieval, VistaGUI decomposes user input into parallel intents, transforming ambiguous commands into specific and verifiable goals. It then retrieves complementary knowledge from offline logs, online sources, and prior interaction history, grounding retrieval in both context and multiple perspectives. This intent-aware multi-source retrieval reduces spurious matches and mitigates the "partial or misleading context" problem, yielding coherent multimodal plans with greater flexibility and robustness.

With a robust plan grounded in accurate knowledge, the next challenge is reliable execution in a dynamic GUI. To this end, VistaGUI integrates structured signals (e.g., APIs) with unstructured modalities (visual perception and OCR) in a unified pipeline. This design directly addresses the second limitation, i.e., brittleness, of prior approaches: stable structural cues protect against superficial changes (e.g., icon color), while unstructured signals capture layout cues and custom components missed by APIs. By fusing the strengths of both, VistaGUI overcomes the "distorted and incomplete" perception bottleneck, achieving a comprehensive understanding of GUI states.

Recognizing that even perfect perception and planning cannot prevent all errors, our final contribution introduces a proactive replanning and resilient recovery mechanism. It systematically records and analyzes past interaction trajectories to support robust replanning and recovery, directly solving the third limitation. When failures occur—such as a missing window—VistaGUI does not discard its plan. Instead, it engages in a Rollback–Correct–Resume cycle: it rolls back to the last valid state using its interaction history, applies a targeted correction by reusing successful past actions, and then resumes execution without redundant replanning. This history-aware recovery ensures robustness with minimal overhead, differentiating VistaGUI from methods that fail or restart unnecessarily.

Together, these three phases form a continuous cycle of planning, acting, and reflecting, enabling VistaGUI to dynamically adapt and execute complex GUI tasks with robustness, efficiency, and compositional reasoning. Extensive experimental results demonstrate the superiority of our VistaGUI over state-of-the-arts. For example, our VistaGUI outperforms the previous SoTA by 11.43% and 8.1% on the WindowsArena (WAA) and OSworld-W datasets, respectively.

## 2 RELATED WORK

**Agent Benchmarks & Architectures.** Benchmarks for computer-operation agents have progressed from offline evaluations with fixed action sequences to online environments that support realistic functional execution. Recent examples include OSWorld Xie et al. (2024) and WindowsAgentArena Bonatti et al. (2024) for desktop tasks, AndroidWorld Rawles et al. (2025) for mobile platforms, and ScreenSpot-Pro Li et al. (2025) for visual grounding. Accordingly, research works have crystallized two main architectural paradigms: monolithic and hierarchical designs.

Monolithic end-to-end agents unify perception, reasoning, and action execution in a single framework, aiming for seamless generalization across interfaces. Systems like UI-TARS Qin et al. (2025a) and AGUVIS Xu et al. (2024) exemplify this approach. Despite strong integration ability, they face steep data requirements and must balance specialization with general reasoning, limiting scalability.

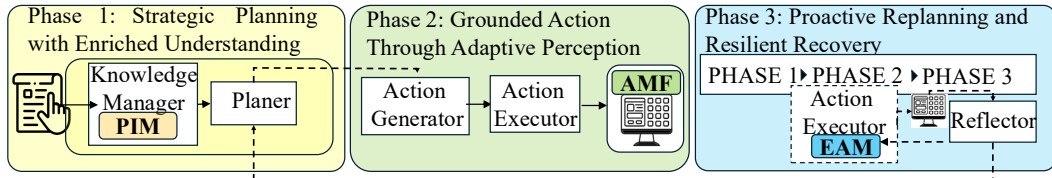

Figure 1: The proactive execution loop of VistaGUI. The five core modules (Planner, etc.) operate across three phases: 1) Strategic Planning, where user intent is enriched with knowledge; 2) Grounded Action, where perception guides execution; and 3) Resilient Recovery, where the Reflector provides feedback for dynamic replanning.

Hierarchical and modular agents address these challenges by decoupling high-level planning from low-level execution, often using Planner–Grounder or Manager–Worker configurations. SeeClick and OS-Atlas Cheng et al. (2024); Wu et al. (2024) exemplify this design, improving flexibility and interpretability. Enhancements include MLLM-based validation, as in GTA-1 Yang et al. (2025), which employs multi-candidate sampling with an MLLM judge to improve robustness on visually complex interfaces. This modular philosophy extends beyond GUI tasks: frameworks such as AutoGen and Agent-S/S2 (Wu et al., 2023; Agashe et al., 2025a;b) provide reusable infrastructures for orchestrating multi-agent collaboration and dynamic tool use. However, modularity can also introduce communication overhead and coordination bottlenecks.

Beyond GUI-specific paradigms, a complementary direction explores agents that dynamically compose tools and APIs to expand capabilities at runtime. Systems such as UFO-2 (Zhang et al., 2025), PyVision (Zhao et al., 2025), and ALITA (Qiu et al., 2025) exemplify this principle, highlighting adaptive tool invocation as a key enabler of flexible and context-aware behaviors.

Our VistaGUI adopts a hierarchical and modular architecture to leverage its flexibility. Unlike prior modular agents that often suffer from brittle perception and simplistic failure handling, VistaGUI introduces dedicated mechanisms for adaptive perception and resilient recovery, directly addressing these critical gaps.

**Knowledge Grounding with RAG.** To enhance reasoning and handle complex tasks, many agents integrate external knowledge via Retrieval-Augmented Generation (RAG). This allows them to retrieve information from user manuals, past successful interaction trajectories, or task-specific documentation to inform their planning process (Kim et al., 2024), state-aware guidelines (Fu et al., 2024), and past experiences (Kagaya et al., 2024). However, the effectiveness of RAG is highly dependent on the quality of retrieved context. Generic retrieval often fails to capture the causal or procedural structure of GUI tasks, which can lead to planning errors.

To overcome this limitation, VistaGUI's strategic planning module moves beyond generic retrieval. By decomposing user intent and employing multi-source retrieval, it generates rich, actionable context, directly tackling the issue of retrieval quality that plagues conventional RAG-based agents.

## 3 METHODOLOGY

### 3.1 OVERVIEW: CORE ARCHITECTURE AND PROACTIVE MODULAR EXECUTION FLOW

VistaGUI is a modular framework designed to overcome key challenges in GUI automation, including imprecise knowledge retrieval, limited perception of the environment, and the absence of structured failure recovery. The overall architecture of VistaGUI is presented in Fig. 1. At its core lies a proactive execution loop that coordinates five specialized modules—the `Planner`, `Knowledge Manager`, `Action Generator`, `Action Executor`, and `Reflector`—to translate high-level user instructions into reliable GUI interactions. This entire process unfolds across three tightly connected phases of reasoning and execution.

**Phase 1: Strategic Planning with Enriched Understanding.** Given the user's input instruction, the process begins as the `Knowledge Manager` enriches the `Planner` with relevant contexts from offline historical logs or online sources, enabling it to decompose the user's input instruction

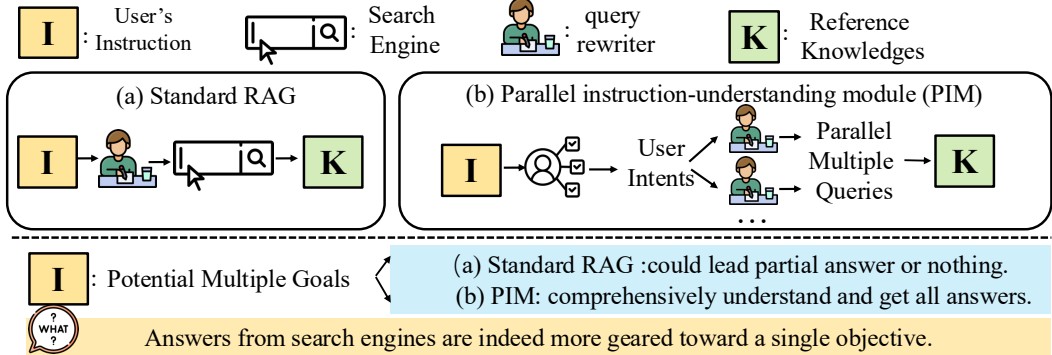

Figure 2: Comparison between Standard RAG and our proposed PIM. (a) Standard RAG's linear process can miss nuances in multi-goal instructions. (b) Our PIM deconstructs the instruction, processes intents in parallel, and fuses knowledge for a more comprehensive understanding.

into a sequence of coherent high-level goals. This integration ensures that planning is not only goal-driven but also grounded in relevant prior knowledge.

**Phase 2: Grounded Action through Adaptive Perception.** Each high-level goal is then refined by the `Action Generator` into executable commands. The `Action Executor`, assisted by adaptive perception, anchors these commands to the current GUI state, ensuring accurate and robust interaction with interface elements.

**Phase 3: Proactive Replanning and Resilient Recovery.** Following execution, the `Reflector` evaluates the outcome and passes feedback to the `Planner`. This closes the loop by enabling proactive adjustments: if progress is on track, the plan is refined and continued; if failure occurs, the system leverages prior context to recover without restarting the entire task.

Together, these phases form a continuous cycle of planning, acting, and reflecting, allowing VistaGUI to adapt dynamically and execute complex GUI tasks with robustness and compositional reasoning. The next sections detail the core innovations within each phase.

### 3.2 PHASE 1: STRATEGIC PLANNING WITH ENRICHED UNDERSTANDING

The execution loop begins with the user's instruction $I$, which sets the stage for high-level reasoning. At this stage, the `Planner`—the system's central strategist—decomposes the instruction into a sequence of subgoals, Plan $= \{g_1, g_2, \ldots, g_n\}$, that guide the agent toward task completion. However, producing an effective plan requires more than syntactic parsing: it demands a deep context-rich understanding of both the instruction and the surrounding environment.

To meet this challenge, VistaGUI introduces the `Knowledge Manager`. Standard Retrieval-Augmented Generation (RAG) often falters on complex instructions with multiple intents, leading to noisy or incomplete knowledge retrieval. To overcome this limitation, our `Knowledge Manager` incorporates a **Parallel Instruction-understanding Module (PIM)** as shown in Fig. 2. The PIM operates in a multi-step process. First, it leverages a large language model prompted to identify and separate distinct user intents. Then, rather than performing a single broad search, it queries external knowledge sources (e.g., web search) for each intent in parallel. Finally, the retrieved information for each intent is concatenated and passed to another LLM call for summarization and fusion into a coherent context. This method not only increases retrieval precision but also ensures that no critical sub-intent is overlooked.

Specifically, this process, represented as $K(I) \rightarrow \{H_{\text{relevant}}, W_{\text{relevant}}\}$, gathers two critical types of information. 1) **Internal Experiences ($H_{\textbf{relevant}}$):** The module accesses a repository of the agent's own past interaction trajectories. It retrieves historical logs that are semantically similar to the current task, allowing the `Planner` to leverage proven strategies and avoid repeating past mistakes. 2) **External Knowledge ($W_{\textbf{relevant}}$):** The module is integrated with a web search engine to query for real-time information. This provides up-to-date, real-world facts necessary to understand the query's context, such as finding a specific address or looking up product details.

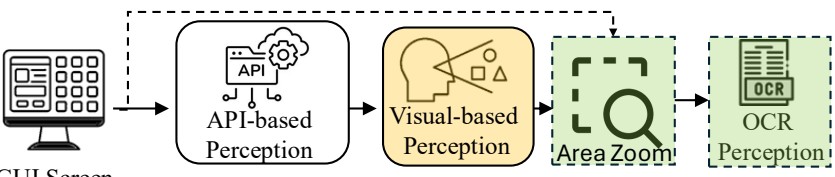

GUI Screen      Adaptive Multi-modal sensing Framework (AMF)

Figure 3: Adaptive Multi-Modal Sensing Module (AMF) first leverages API-based Perception for a quick structural overview, then applies Visual-based Perception for local refinement. For areas unparsable by the API, it performs an 'Area Zoom' and invokes OCR to recognize key text.

Armed with this enriched context, the `Planner` generates a well-informed and contingency sequence of subgoals. By analyzing the key entities and intents from the instruction alongside the data from the `Knowledge Manager`, it formulates each foundational step toward the solution. Crucially, this plan is multi-modal, composed of subgoals that extend beyond direct GUI manipulation. The `Planner` can generate commands for on-screen interactions or delegate tasks to auxiliary tools, such as web and document extraction for information gathering or Python/shell execution for computation and system-level commands. This hybrid approach enables VistaGUI to tackle complex tasks that require a fluid combination of on-screen actions and external data processing.

Furthermore, this plan is not static. The `Planner` operates within a continuous feedback loop, guided by the `Reflector` in Phase 2 (see Fig. 1). After each subgoal is executed, the outcome is evaluated, and the plan is refined in real time. This proactive replanning ensures that agent adapts to unexpected changes, recovers from errors, and incrementally converges toward success.

Accordingly, Phase 1 establishes a novel planning paradigm where PIM's parallel intent decomposition and dual knowledge integration transform raw instructions into adaptive multi-modal plans, advancing beyond traditional RAG by equipping agents with the contextual depth and flexibility.

### 3.3 PHASE 2: GROUNDED ACTION THROUGH ADAPTIVE PERCEPTION

With Plan $= \{g_1, g_2, \ldots, g_n\}$ in Phase 1, VistaGUI transitions from strategic planning to concrete execution. The `Action Generator` translates each high-level subgoal in $\{g_1, g_2, \ldots, g_n\}$ into an executable low-level action, typically a code snippet targeting a specific UI element.

Successful execution of this action hinges on the agent's ability to accurately perceive the current screen state, $S_t$. This critical task is handled by the `Action Executor`, which relies on our **Adaptive Multi-Modal Sensing Module (AMF)**, illustrated in Fig. 3. The core challenge in GUI perception is the heterogeneity of UI elements: some are well-structured entities exposed via accessibility APIs, while others are custom images, non-standard controls, or embedded content.

To address this, our AMF employs an efficient two-stage strategy to build a comprehensive understanding of the active window ($W_{\text{active}}$), which is the primary focus of user interaction. The final set of all perceived elements, $E$, is formed by the union of outputs from a primary and a fallback perception stage:

$$E = \Pi(W_{\text{active}}) = \Pi_{\text{API}}(W_{\text{active}}) \cup \Pi_{\text{fallback}}(W_{\text{context}}). \tag{1}$$

Here, $W_{\text{context}}$ is the region within the active window that remains unparsed by our primary method. We detail the two stages below.

1) **Primary Perception Stage ($\Pi_{\textbf{API}}$):** This stage prioritizes reliability and semantic richness by leveraging the accessibility tree, which provides structured details like element roles (e.g., button, text field), names, and values. We first deploy an API-based expert ($\Phi_{\text{API}}$) across the entire active window. This expert is the most efficient and accurate method for gathering structured element data.

$$\Pi_{\text{API}}(W_{\text{active}}) = \Phi_{\text{API}}(W_{\text{active}}) \tag{2}$$

2) **Fallback Perception Stage ($\Pi_{\textbf{fallback}}$):** Many UI elements, such as custom-rendered controls, video players, or proprietary interfaces, are not exposed through the accessibility tree, creating "blind spots." To achieve complete perception, our fallback stage intelligently fills these gaps.

This stage operates exclusively on the contextual region ($W_{\text{context}}$), defined as the area within the active window that the API-based expert could not analyze. To perceive elements in this region, we

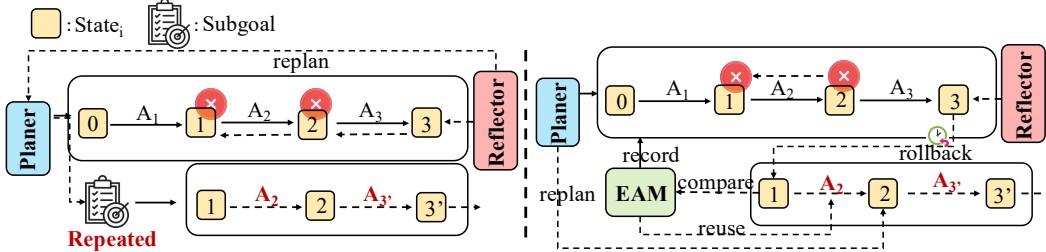

Figure 4: Our checkpoint-based recovery mechanism. A standard agent (left) must restart an entire subgoal upon failure. Our EAM (right) rolls back to the last successful checkpoint for efficient and resilient replanning. In the fig, $A_i$ denotes the $i$-th action.

employ a Visual Perception expert ($\Phi_{\text{VP}}$) and an OCR expert ($\Phi_{\text{OCR}}$) in tandem:

$$\Pi_{\text{fallback}}(W_{\text{context}}) = \Phi_{\text{VP}}(W_{\text{context}}) \cup \Phi_{\text{OCR}}(W_{\text{context}}). \tag{3}$$

This dual-stage approach enables adaptive perception that unifies structured and unstructured UI understanding, capturing elements missed by the API—whether graphical components or rendered text—and addressing a key bottleneck in GUI automation. Moreover, by applying these more resource-intensive methods only to unparsed regions, the system remains efficient while maintaining comprehensive coverage, ensuring reliable operation across diverse real-world interfaces where traditional API-only methods would fail. Finally, the `Action Executor` uses the complete and precise set of elements $E$ to ground the generated action code, enabling robust interactions like button clicks or text entry in a specific field. As for the details of all experts, we show them in the appendix A.3.

### 3.4 PHASE 3: PROACTIVE REPLANNING AND RESILIENT RECOVERY

After executing each action, VistaGUI enters a critical assessment and adaptation phase, driven by the `Reflector`. This phase embodies the framework's proactive and resilient nature, enabling dynamic adjustment to successes and failures with specialized strategies.

For successful actions, the resulting state-action pair is stored as a validated checkpoint in the `EAM`. Rather than blindly proceeding to the next step in a static plan, the `Planner` re-evaluates the new GUI screen state, the full interaction history, and the remaining subgoals. This proactive replanning allows the agent to adapt to dynamic or unexpected changes in the GUI, ensuring that all subsequent steps remain optimal.

Conversely, if an action fails, VistaGUI avoids the costly inefficiency of restarting the entire task from scratch. It instead initiates a robust recovery process managed by our **Environment-Aware State Management (EAM)** module. As shown in Fig. 4, a standard agent (left) is forced to discard all progress within a sub-goal upon failure, triggering a full replan of the entire action sequence. In contrast, our EAM-driven agent (right) leverages its recorded state history to roll back only to the last successful checkpoint, thereby salvaging prior work and enabling a more efficient, localized recovery. This checkpoint-based approach follows a three-stage protocol:

1) **Trajectory Logging and Checkpointing:** Each executed action $a_t$ and its resulting state $S_t$ are recorded as part of a trajectory. After each action $a_t$, the `Reflector` provides an outcome $r_t \in \{\text{success}, \text{failure}\}$. If $r_t = \text{success}$, the state transition $(S_{t-1}, a_t, S_t)$ is stored as a validated checkpoint, $C_t$. The agent's history, $H$, is an ordered sequence of these checkpoints:

$$H = \{C_1, C_2, \ldots, C_k\}, \tag{4}$$

where $k$ is the index of the last successfully executed step.

2) **Failure Detection and Trajectory Invalidation:** When the `Reflector` detects a failure at timestep $t$ ($r_t = \text{failure}$), the EAM immediately identifies the last validated checkpoint, $C_k$, from history $H$, and marks the intervening actions taken from state $S_k$ to the point of failure as the invalid trajectory segment, $T_{k \to t}^{\text{invalid}}$:

$$T_{k \to t}^{\text{invalid}} = ((S_k, a_{k+1}), (S_{k+1}, a_{k+2}), \ldots, (S_{t-1}, a_t)). \tag{5}$$

This records the failed path from the state $S_k$, and prevents the agent from repeating known errors.

3) **Automated Rollback and Constrained Replanning:** Upon identifying an invalid trajectory $T_{k \to t}^{\text{invalid}}$, the EAM's recovery protocol begins by rolling back to the last known-good state, $S_k$. This action-based restoration involves matching the failed state against historical states. If a match is successful, the EAM retrieves and re-executes the corresponding action sequence $(a_j, \ldots, a_k)$ from history, deterministically returning the agent to $S_k$ (where $a_j$ is the action of the matched state). Subsequently, the 'Planner' is instructed to generate a new path constrained to avoid the failed action sequence in $T_{k \to t}^{\text{invalid}}$. This prevents the agent from retrying a known failed strategy and encourages the exploration of new solutions. For the detailed algorithm, please refer to Appendix A.5.

## 4 EXPERIMENTS

**Benchmarks.** We evaluate our agent's performance on two rigorous benchmarks designed for Windows-based automation. Our primary testbed is the **Windows Agent Arena (WAA)** (Bonatti et al., 2024), which features 154 real-world tasks spanning 15 common Windows applications, including office suites and web browsers. To further test our agent's capabilities on core productivity operations, we also evaluate on **OSWorld-W** (Xie et al., 2024), a Windows-specific subset of the OSWorld benchmark. This benchmark consists of 49 tasks focused on office applications, browser interaction, and file system management. A key advantage of both benchmarks is their reliance on automated verification scripts to programmatically check for task completion, ensuring an objective and reproducible evaluation.

**Baselines.** We compare `VistaGUI` with five representative state-of-the-art Computer User Agents (CUAs), each leveraging a powerful foundation model like GPT-4o as its core inference engine. The detailed descriptions for the compared methods are given in the appendix. These baselines were selected to represent a diverse range of architectural paradigms in GUI automation (e.g., single-agent vs. multi-agent, hybrid-input vs. vision-only). To ensure a fair and practical comparison, each agent is restricted to a maximum of 50 execution steps per task, preventing excessively long or inefficient attempts. More details of baseline methods are given in the appendix A.6.

**Implementation Details.** For our **Adaptive Multi-Modal Sensing Framework (AMF)**, `VistaGUI` integrates an icon-text detector OmniParser for visual perception, **Paddle OCR** for textual data extraction, and queries to accessibility APIs (e.g., DOM trees for web, UI Automation for desktop) for structural information. The backbone large language models evaluated for planning and reasoning using **GPT-4o** for fair comparison.

**Evaluation Metrics.** We utilize the primary metric standard in GUI automation benchmarks for performance evaluation: **Success Rate (SR):** Defined as the percentage of tasks successfully completed. Task completion is validated automatically by the official verification scripts provided by the **WAA** and **OSWorld-W** benchmarks, ensuring an consistent measurement of performance.

### 4.1 QUANTITATIVE COMPARISON

As shown in Table 1, VISTAGUI achieves state-of-the-art (SOTA) results on both the WAA and OSWorld-W benchmarks, significantly outperforming all prior methods.. Specifically, it achieves a success rate of 32.2% on WAA and 22.4% on OSWorld-W. This performance far exceeds all comparative methods, representing a significant absolute improvement of 11.43% and 8.1% over the best-performing baseline, Operator (20.8% on WAA, 14.3% on OSWorld-W), respectively.

Table 1: Comparison of SR (%) across agents on WAA and OSWorld-W benchmarks.

| Agent | Model | WAA | OSWorld-W |
|---|---|---|---|
| UFO | GPT-4o | 19.5 | 12.2 |
| NAVI | GPT-4o | 13.3 | 10.2 |
| OmniAgent | GPT-4o | 19.5 | 8.2 |
| Agent S | GPT-4o | 18.2 | 12.2 |
| Operator | computer-use | 20.8 | 14.3 |
| VISTAGUI | GPT-4o | 32.23 | 22.4 |

In the breakdown of application types on the WAA benchmark (Table 2), VISTAGUI's advantage is even more pronounced. It demonstrates absolute dominance in technically demanding tasks such as 'Coding' (63.70%) and 'Windows System' management (60.6%). More importantly, VISTAGUI successfully overcomes the bottleneck that existing methods face with 'Office' software, achieving a 7.8% success rate where all other GPT-4o-based agents scored 0.0%, showcasing

Table 2: SR (%) breakdown by application type on WAA and OSWorld-W.

| Agent | Model | WAA | | | | | | OSWorld-W | |
| | | Office | Web Browser | Windows System | Coding | Media & Video | Windows Utils | Office | Cross-App |
|---|---|---|---|---|---|---|---|---|---|
| UFO | GPT-4o | 0.0 | 23.3 | 33.3 | 29.2 | 33.3 | 8.3 | 18.5 | 4.5 |
| NAVI | GPT-4o | 0.0 | 20.0 | 29.2 | 9.1 | 25.3 | 0.0 | 18.5 | 0.0 |
| OmniAgent | GPT-4o | 0.0 | 27.3 | 33.3 | 27.3 | 30.3 | 8.3 | 14.8 | 0.0 |
| Agent S | GPT-4o | 0.0 | 13.3 | 45.8 | 29.2 | 19.1 | 22.2 | 22.2 | 0.0 |
| Operator | computer-use | 7.0 | 26.7 | 29.2 | 29.2 | 28.6 | 8.3 | 22.2 | 4.5 |
| VISTAGUI | GPT-4o | 7.8 | 40.5 | 60.6 | 63.7 | 40.2 | 44.4 | 33.3 | 9.1 |

Table 3: Ablation study on PIM (%).

| Method | SR | Ret. Acc |
|---|---|---|
| Baseline | 15.6 | - |
| VISTAGUI | 22.4 | 90.3 |
| VISTAGUI w/o WS | 16.3 | 71.5 |

Table 4: Ablation study on EAM (%).

| Method | SR | Fail. Rec. |
|---|---|---|
| VISTAGUI | 22.4 | 79.4 |
| w/o Recovery | 19.6 | 65.3 |
| w/o Replanning | 18.4 | 10.4 |

its unique capability in handling complex GUI interactions. Similarly, on the OSWorld-W benchmark, VISTAGUI continues its leading performance, especially in 'Office' (33.3%) and 'Cross-App' (9.1%) tasks, where its success rates far surpass those of its competitors. This indicates that VISTAGUI can not only efficiently complete tasks within a single application but also possesses the powerful ability to understand and execute complex cross-application workflows.

## 4.2 ABLATION STUDY

**Impact of Instruction Understanding (PIM).** To validate our Parallel Instruction-understanding Module (PIM), we conducted an ablation study comparing three retrieval methods in Table 3: 1) Baseline with no web retrieval, 2) WS is the variant with direct retrieval on raw instructions, and 3) our VISTAGUI model using PIM for intent-based parallel retrieval. Retrieval Accuracy (Ret. Acc) is also introduced in the experiments. This metric is the percentage of instances where the retrieved knowledge is judged to be successful by an automated evaluator, GPT-4o, based on its relevance, accuracy, and sufficiency in addressing the user's query. A retrieval is deemed successful if and only if the returned information is comprehensive enough to cover all of the user's intents as expressed in their original instruction. This metric clearly reveals the limitations of simpler approaches. The WS-based model, constrained by ambiguous raw instructions, often retrieved irrelevant information and thus achieved a low Ret. Acc. In stark contrast, our model achieved the highest accuracy by employing PIM's 'understand-first, retrieve-second' strategy. By decomposing ambiguous instructions into clear, parallel intents before performing a contextualized retrieval, it consistently fetched relevant knowledge. This demonstrates that a sophisticated retrieval module capable of deep instruction understanding is essential for high performance and, ultimately, high success rates for the GUI agent.

**Impact of Adaptive Sensing (AMF).** In Fig. 5, we conducted an ablation study comparing our full model against two variants: (1) an 'API-based' version that relies purely on API perception data, and (2) a 'Visual-based' version that uses only visual perception data. As shown in the performance comparison figure, our integrated approach significantly outperforms both ablated variants across both benchmarks. On the WAA benchmark, our model achieves a 32.2% success rate, outperforming the 'API-based' (28.3%) and 'Visual-based' (25.7%) versions by 3.9% and 6.5%, respectively. Similarly, on the OSWorld-W benchmark, our model leads with a 22.4% success rate, compared to 20.4% and 16.3% for the 'API-based'

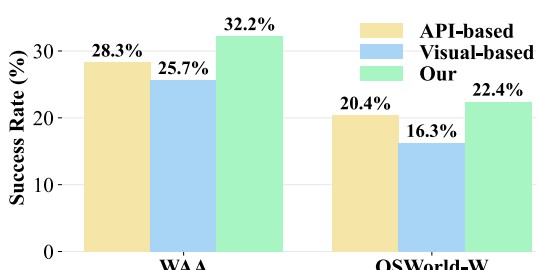

Figure 5: Ablation study on Adaptive Multi-Modal Sensing Framework (AMF).

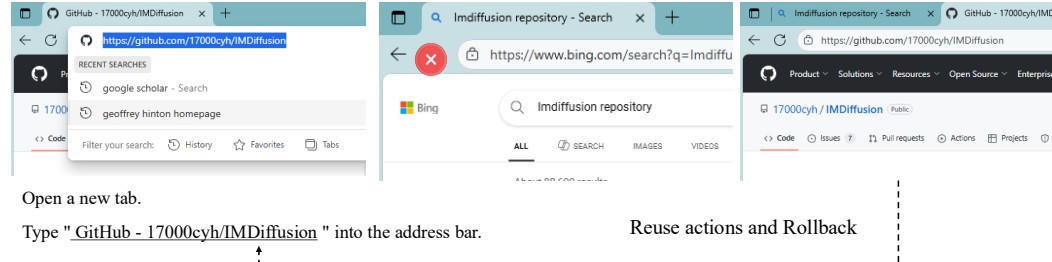

Open a new tab.

Type " GitHub - 17000cyh/IMDiffusion " into the address bar.

Reuse actions and Rollback

Figure 6: An example of VistaGUI's recovery protocol. The agent recovers from a browser error where a URL was misinterpreted as a search query by rolling back and re-issuing the correct navigation command.

and 'Visual-based' versions. These results provide strong evidence that adaptively switching different perceptions is crucial for robust GUI navigation and interaction, validating the necessity of each component in our multi-modal design.

**Impact of Failure Recovery (EAM).** We further investigate the contributions of our key components: the recovery mechanism and constrained replanning. To do this, we first define the Failure Recovery Rate (FRR) as the success rate of the recovery mechanism itself, measured as the percentage of failures after which a successful rollback to a prior state was achieved. We compare our full model against two ablated variants: 1) w/o Recovery Mechanism, which attempts to replan from the failure state without rolling back, and 2) w/o Constrained Replanning, which rolls back but then enters an execution loop by bypassing the action generator. The results in Table. 4 reveal their critical roles. Disabling the recovery mechanism entirely (w/o Recovery Mechanism) causes the SR to drop from 22.4% to 19.6% and, more dramatically, reduces the FRR from a robust 79.4% to a 65.3%. Similarly, removing constrained replanning (w/o Constrained Replanning) decreases the SR to 18.4% and causes the FRR to plummet to 10.4%. These findings underscore that both the ability to roll back to a safe state and the intelligence to replan from it are indispensable for agent resilience.

### 4.3 VISUALIZATION

Fig. 6 provides a concrete example of our agent's resilient recovery capability in a scenario that commonly derails less robust agents. In this web-based task, the agent's initial navigation attempt is erroneously redirected to a search engine—a frequent and unpredictable challenge in real-world web environments. While many agents would either fail completely or get stuck in a costly replanning loop from the unexpected search results page, VistaGUI demonstrates its robustness. Its Environment-Aware State Management (EAM) module correctly identifies the deviation from the expected outcome, programmatically rolls back the single failed step, and re-issues a more precise navigation command to reach the correct destination. This case study was specifically conducted on the web to prove that VistaGUI's recovery protocol is not limited to stable desktop applications but is effective even in the most dynamic and unpredictable conditions. For additional case studies, please refer to the appendix A.7.

## 5 CONCLUSION

In this work, we presented VistaGUI, a robust and compositional agent that tackles three core challenges in GUI automation: inaccurate knowledge retrieval, fragile single-modality perception, and weak failure recovery. VistaGUI advances the state of the art through three key innovations—the Parallel Instruction-understanding Module (PIM), the Adaptive Multi-modal Sensing Framework (AMF), and the Environment-aware State Management (EAM)—which together deliver higher task success rates, faster recovery, and stronger robustness compared to competitive baselines. This establishes VistaGUI as a significant step toward general-purpose, reliable agents for real-world computer interaction, moving beyond the limitations of prior reactive systems.

**Future Work, Broader impacts & Reproducibility.** Future directions include extending long-horizon strategic planning, enabling proactive user queries to resolve ambiguity, and improving computational efficiency for lightweight deployment in resource-constrained environments. See the discussion of broader impacts and reproducibility details in subsection A.1 and A.2.

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
