# A  APPENDIX

## A.1  ETHICS STATEMENT

The Large Language Models (LLMs) powering VistaGUI may reflect biases from their training data. As an autonomous agent designed to interact with real-world graphical user interfaces, there is a potential risk of performing unintended actions, accessing sensitive information, or processing harmful content retrieved from external web sources. This work is intended solely for academic and scientific purposes to advance the understanding of autonomous agents. Any actions or outputs generated by the agent do not represent the views of the authors. We are committed to the responsible development of AI technologies that are safe, controllable, and aligned with societal values.

## A.2  REPRODUCIBILITY STATEMENT

We detail the architecture and workflow of VistaGUI in the Methods section. Full implementation details for our core modules (PIM, AMF, and EAM), along with hyperparameters and the experimental setup, are described in Section 4 and the appendix. The source code and evaluation benchmarks will be made publicly available to ensure reproducibility and facilitate future research.

## A.3  PERCEPTION MODULES AND REGION DEFINITIONS.

**API-based Expert** ($\Phi_{\text{API}}$)**:** This is a specialized function that takes a window region $W$ and returns a set of structured elements, $E_{\text{API}}$, by **parsing the window's accessibility tree**. This method is the most reliable perception source as it directly retrieves rich semantic information such as element types, positions, and contents.

**Visual Perception Expert** ($\Phi_{\text{vp}}$)**:** This expert is powered by a **icon-text detection**, such as Omniparse. It takes an image of a region $W$ as input and returns a set of visually grounded elements, $E_{\text{vp}}$. By analyzing pixels, this expert can identify and understand UI elements in complex or non-standard interfaces.

**OCR-based Expert** ($\Phi_{\text{OCR}}$)**:** This is a function that uses **Optical Character Recognition (OCR) technology**. It takes a region $W$ and returns a set of textual elements, $E_{\text{OCR}}$, along with their coordinates. This expert serves as a fallback option when other methods fail to provide text information.

**Active Window Region** ($W_{\text{active}}$)**:** This is the spatial area on the screen occupied by the current active window. Its position and size are determined by $\Phi_{\text{API}}$. This region typically represents the window the user is currently interacting with.

**Contextual Region** ($W_{\text{context}}$)**:** This region is defined as the area of other elements in the current window that are not perceived by the API-based Expert. This region includes elements like custom-rendered controls, video players, or game interfaces that do not provide an accessibility tree.

## A.4  PERCEPTION COMPARISON.

Fig. 7 visually illustrates the vast difference in the quality of UI component understanding between our unified perception method (API+Visual) and a singular OCR-based approach. The output on the left shows that our unified method not only successfully identifies the three actionable options on the interface ("Inspect Document," "Check Accessibility," and "Check Compatibility") but also precisely captures their structured information, including their respective icons, titles, and descriptive text. This multi-modal perception understands that each component is a distinct, functional entity.

In contrast, the pure OCR-based perception on the right can only 'flatten' the UI elements into an undifferentiated block of raw text. It completely loses critical structured information such as icons, layout, and element boundaries, failing to distinguish that this is a list of three separate options. For an agent that needs to make decisions and interact with the UI, this loss of information is crippling, as it prevents the formation of a correct understanding of the UI and leads to failures in subsequent planning and execution.

Figure 8 further visualizes the limitations of different perception methods versus the comprehensiveness of our unified approach on a complex web interface. The purely visual-based detection on the

left, while able to identify some graphical blocks with clear boundaries, cannot understand the logical hierarchy of elements and misses a significant number of text and interactive elements that lack distinct visual borders. The purely API-based detection in the middle can capture the logical structure from the DOM tree, but it often cannot access custom-rendered graphical components, and its provided bounding boxes may deviate from the actual visual layout. Both of these single-modality methods can only provide a partial, or even distorted, understanding of the interface.

In contrast, the detection results from our unified method (OCR+Visual-Based) on the right are far more comprehensive and fine-grained. It successfully combines visual and textual information, not only identifying the main structural blocks but also precisely locating every text label, button, and input field within them. This comprehensive and accurate perceptual ability is the foundation for ensuring an agent can perform reliable reasoning and precise operations in real-world, complex environments.

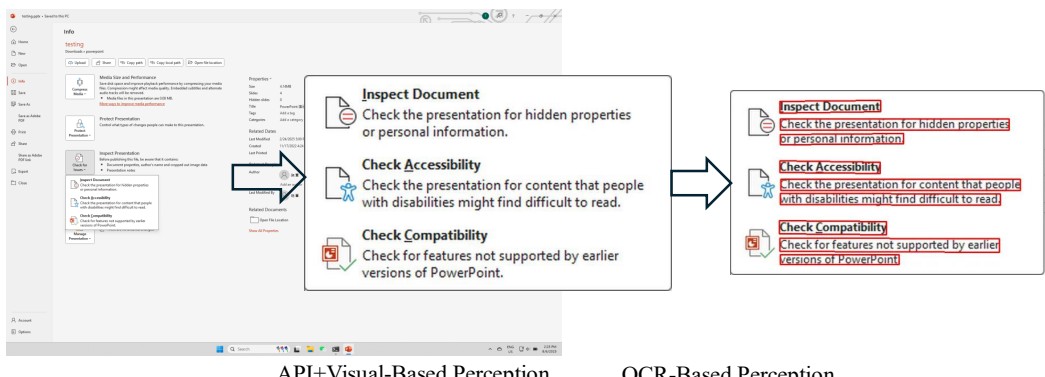

API+Visual-Based Perception       OCR-Based Perception

Figure 7: The show results using different percetions.

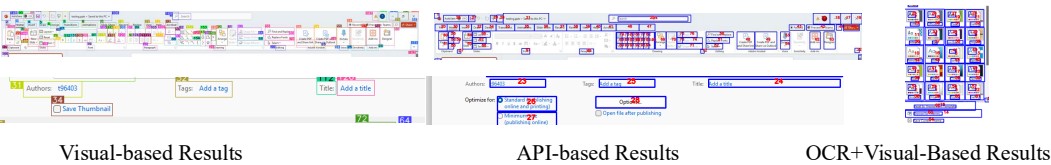

Visual-based Results       API-based Results       OCR+Visual-Based Results

Figure 8: Comparative Visualization of Element Detection Results.

## A.5 Automated Rollback and Constrained Re-planning

**State Restoration (Rollback):** The primary objective of state restoration is to efficiently return the agent to a recent, validated state without performing a computationally expensive full reset from the initial state $S_0$. The process is triggered upon detecting a failure at state $S_t$, which could be a collision, a kinematic constraint violation, or a perception error. At this point, the system must identify a suitable restoration target. To achieve this, the Emergency Autonomy Module (EAM) consults a history buffer, $H$, which maintains a record of the agent's recently traversed states and their corresponding statuses.

Within this history, states that have successfully passed all safety and validity checks are marked as "validated checkpoints." The system performs a reverse search from the failed state $S_t$ through the history $H$ to locate the most recent of these checkpoints, designated as $S_k$. This state is selected as the restoration target because it represents the most recent known-good configuration, a strategy that is crucial for efficiency. It minimizes the amount of progress that must be undone and focuses the recovery effort solely on the problematic segment of the trajectory. Subsequently, a specialized recovery policy, $\pi_{\text{recovery}}$, is employed to generate the actions needed to transition the environment from the failed state $S_t$ back to this target state $S_k$.

**Constrained Replanning:** Following a successful restoration to state $S_k$, the agent must generate an alternative plan to achieve the user's instruction . To achieve this, the planning process is constrained

by the knowledge of the recent failure. Specifically, the planner must generate a new action, $a'_{k+1}$, that diverges from the failed trajectory at the first step. This is formalized by the following constraint:

$$a'_{k+1} = \pi'(S_k, I) \quad \text{s.t.} \quad a'_{k+1} \neq a_{k+1} \tag{6}$$

This simple yet powerful negative constraint on the initial action of the failed segment forces the agent away from its previously unsuccessful path. Consequently, the planner is compelled to explore an alternative strategy from the restored checkpoint $S_k$, effectively bypassing the conditions that led to the original failure and increasing the likelihood of success on the subsequent attempt.

## A.6 BASELINES

We compare our framework with the five representative state-of-the-art methods, each leveraging GPT-4o as the inference engine: **UFO** (Zhang et al., 2025): A pioneering multi-agent framework designed for Windows automation, which integrates UI Automation (UIA) data with visual perception. **NAVI** (Zhao et al., 2025): The official single-agent baseline for the WAA benchmark, utilizing both screenshots and accessibility data for GUI understanding. **OmniAgent** (Lu et al., 2024): An agent that employs an OmniParser for visual grounding combined with a large language model for action planning. **Agent S** (Agashe et al., 2025a): Features a multi-agent architecture with experience-driven hierarchical planning, optimized for complex, multi-step tasks. **Operator** (OpenAI, 2024b): A recent high-performance agent from OpenAI that simulates human-like mouse and keyboard interactions using only screenshots as input.

## A.7 CASE STUDY

Figure 9 demonstrates our agent's ability to successfully execute a foundational multi-step task within the WAA benchmark. The instruction, "Open the Clock app, navigate to the world clock list, and add a clock for Kyoto, Japan's timezone," requires a precise sequence of GUI operations. The agent accurately performs this series by (a) clicking the navigation button, (b) selecting the "World clock" list item, (c) clicking the "Add a new city" button, (d) typing "Kyoto" into the search bar, and (e) selecting the correct item from the dynamically generated search suggestions. This case successfully showcases the agent's core GUI automation capabilities, including understanding and executing sequential instructions, navigating between different screens, performing text input, and interacting with dynamic content, which are fundamental for tackling more complex tasks.

Open the Clock app, navigate to the world clock list, and add a clock for Kyoto, Japan's timezone.

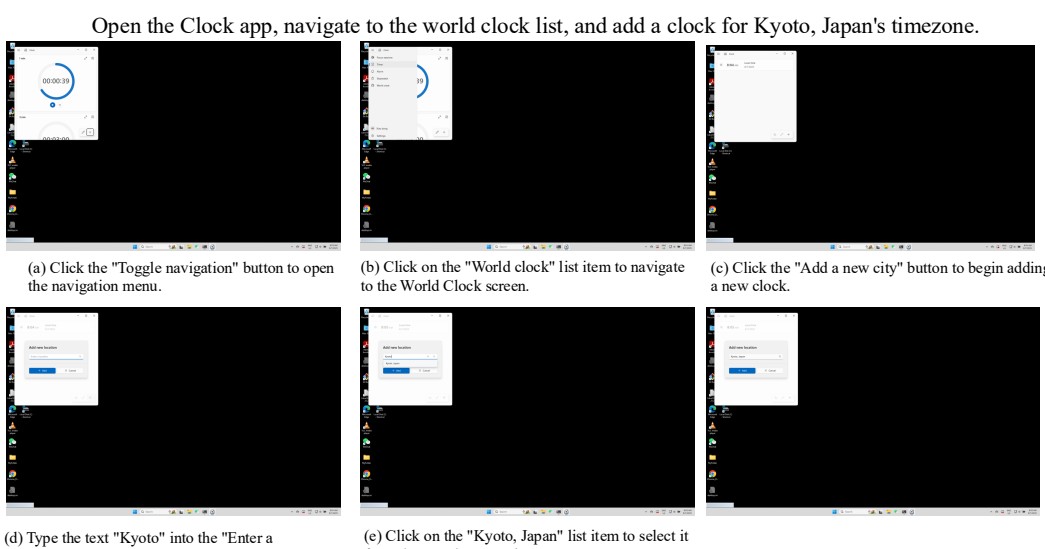

(a) Click the "Toggle navigation" button to open the navigation menu.

(b) Click on the "World clock" list item to navigate to the World Clock screen.

(c) Click the "Add a new city" button to begin adding a new clock.

(d) Type the text "Kyoto" into the "Enter a location" search bar.

(e) Click on the "Kyoto, Japan" list item to select it from the search suggestions.

Figure 9: A successful task of the clock in the WAA benchmark.

In contrast to the foundational task in the previous figure, Figure 10 showcases our agent's capacity to solve a complex, high-level instruction with constraints from the Winbench benchmark. The user's goal, "Help me quickly remove all notes in the slide... without looping through each slide

Help me quickly remove all notes in the slide of the testing ppt, without looping through each slide one-by-one.

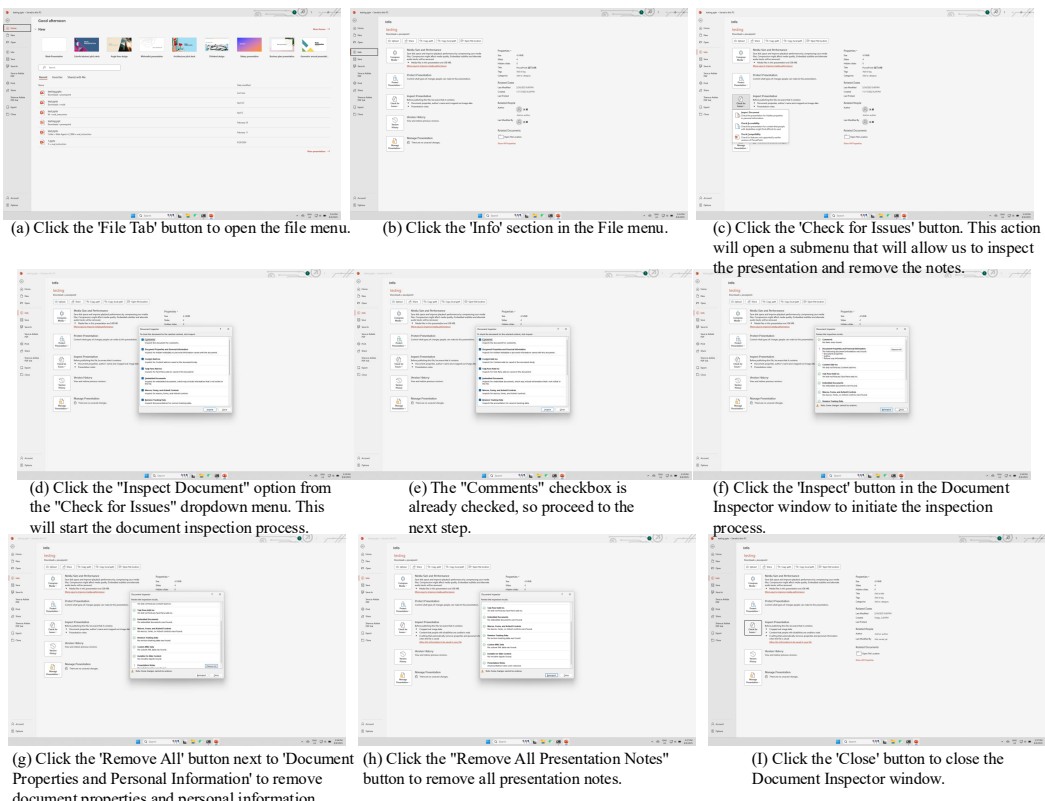

(a) Click the 'File Tab' button to open the file menu.

(b) Click the 'Info' section in the File menu.

(c) Click the 'Check for Issues' button. This action will open a submenu that will allow us to inspect the presentation and remove the notes.

(d) Click the "Inspect Document" option from the "Check for Issues" dropdown menu. This will start the document inspection process.

(e) The "Comments" checkbox is already checked, so proceed to the next step.

(f) Click the 'Inspect' button in the Document Inspector window to initiate the inspection process.

(g) Click the 'Remove All' button next to 'Document Properties and Personal Information' to remove document properties and personal information.

(h) Click the "Remove All Presentation Notes" button to remove all presentation notes.

(I) Click the 'Close' button to close the Document Inspector window.

Figure 10: A successful task of PowerPoint. In the extra benchmark Winbench.

one-by-one," is not a simple command but a problem requiring deep application knowledge for an efficient solution. The agent successfully decomposes this abstract goal into a non-obvious workflow deeply nested within multiple menus (File → Info → Check for Issues → Inspect Document). This case highlights the agent's advanced reasoning and planning capabilities: it correctly interprets the "without looping" constraint to find a global solution, performs deep navigation through a complex application structure, and exhibits contextual awareness by recognizing a pre-checked box in step (e). This demonstrates that our agent functions not merely as an instruction executor, but as a proficient problem-solver that can understand complex intent and devise efficient execution paths.