# OpenReview forum: "VistaGUI: Towards More Robust and Intelligent GUI Automation"
_ICLR.cc/2026/Conference — Submitted to ICLR 2026_

### Official Review · Reviewer_rYRc · 2025-10-20

**Soundness:** 1
**Presentation:** 2
**Contribution:** 1
**Rating:** 2
**Confidence:** 3

**Summary:**

This paper proposes a GUI automation framework, **VistaGUI**, designed to address three claimed limitations:
1. Low retrieval accuracy in RAG
2. Over-reliance on a single modality
3. Inadequate failure recovery mechanisms

VistaGUI tackles these challenges through:
1. A parallel instruction-understanding module to enhance retrieval accuracy
2. A multimodal sensing module that selects the optimal perception modality from API-based queries, visual input, and OCR
3. A recovery mechanism to handle execution failures

These components are organized into three phases and five modules.

**Strengths:**

1. The paper is well-structured: it identifies existing limitations, explains approaches to address them, and provides evaluations to test each proposed method.
2. Results indicate that the proposed approach outperforms baseline methods.

**Weaknesses:**

The paper contains many vague, unqualified, or unsupported statements that can be interpreted in multiple ways, making the text distracting. Examples include:

1. **Line 016:** “...over-reliance on single modality perception...” is vague because the reliance is not quantified. Readers cannot judge what constitutes “over-reliance,” making the claim unconvincing.
2. **Line 210:** “...two critical types...” is vague because “critical” is not justified. Why are these types critical? Are there other non-critical types? Similarly, “...enters a critical phase...” in line 296 lacks justification.
3. **Line 215:** “provides...facts necessary to understand the query's context.” Are all provided facts truly necessary?
4. **Line 305:** “It instead initiates a robust recovery process...” How is the process robust? This is unsupported.
5. **Line 324:** “Automated Rollback...” How is it automated?
6. **Line 410:** “...is comprehensive enough...” How is comprehensiveness measured?

7. For the first limitation, claimed as “low retrieval accuracy in retrieval-augmented generation (RAG)” (line 015), the paper later narrows this to “Standard RAG often falters on complex instructions with multiple intents” (line 202). This shift is problematic because these two limitations are not equivalent. Addressing the latter does not necessarily address the former unless the latter is the sole or primary cause—which is not stated and would be a bold assumption without evidence.

8. The proposed solution for limitation 1—“rather than performing a single broad search, it queries for each intent in parallel... then concatenates the retrieved information and passes it to another LLM for summarization” (lines 205–208)—seems trivial. Is there evidence that LLMs (e.g., GPT-4o used in the paper), when performing a broad search, do not implicitly break down queries into subgoals? Also, concatenating and summarizing for all subgoals seems unconvincing because each subgoal ends up with the same summarized information.

9. Phase 2, intended to address limitation 2, again shifts the problem from “over-reliance on a single modality” to “The core challenge in GUI perception is the heterogeneity of UI” (line 240). It is unclear how the single-modality issue is resolved.

10. The proposed approach for limitation 3—a checkpoint mechanism—also appears trivial.

11. Finally, the phrase “proven strategies” in line 213 is unclear. What strategies? Proven by what evidence?

**Questions:**

1. In line 240, what does the “F” in “AMF” stand for in “Adaptive MultiModal Sensing Module”?
2. In the experiments, does each task (e.g., the 154 tasks mentioned in line 337) include multiple subgoals?
3. Is this the first GUI agent approach to break commands into subgoals and retrieve related information separately?
4. Is this the first GUI agent approach to leverage multiple modalities, while prior work uses only one (OCR or API)?
5. Is this the first GUI agent approach to implement a checkpoint recovery mechanism?

---

### Official Review · Reviewer_XhzL · 2025-10-28

**Soundness:** 2
**Presentation:** 2
**Contribution:** 1
**Rating:** 4
**Confidence:** 4

**Summary:**

The paper introduces VistaGUI, a compositional agent framework for GUI automation. The work claims to address three challenges in this domain: (1) inaccurate knowledge retrieval, (2) fragile single-modality perception, and (3) weak failure recovery. To tackle these, the authors propose three components: the Parallel Instruction-understanding Module (PIM) for task parsing, the Adaptive Multi-modal Sensing Framework (AMF) for robust perception, and the Environment-aware State Management (EAM) for recovery and adaptation.

The system is structured around a continuous cycle of planning, acting, and reflecting, with the aim of improving robustness, efficiency, and compositional reasoning. Empirical evaluations are reported on benchmark datasets such as WindowsArena (WAA) and OSworld-W, where VistaGUI achieves performance improvements over prior baselines.

**Strengths:**

1. The paper demonstrates originality by addressing GUI automation from a robustness and intelligence perspective, which is relatively underexplored compared to traditional automation approaches.

2. The proposed VistaGUI framework effectively combines vision-based techniques with reasoning components, allowing the system to adapt to diverse and dynamic GUI environments. The technical quality is strong, with a well-structured methodology and clear problem formulation.

3. The experiments are relatively comprehensive, covering multiple application domains, and convincingly show the framework’s improvements over baseline methods.

4. Clarity is another strength, as the paper is well-organized and the figures/tables clearly illustrate the system architecture and results. The significance lies in its potential impact on practical GUI automation, making automated systems more reliable and scalable for real-world deployment.

**Weaknesses:**

1. The framework integrates vision and reasoning for GUI automation, but the overall approach is incremental compared to prior vision-based automation systems. The contribution is more in combining existing components than in introducing fundamentally new concepts.

2. The evaluation is broad but lacks depth in some critical scenarios. For instance, large-scale stress testing, real-time responsiveness, and robustness under highly dynamic or noisy environments are insufficiently explored.

3. The adaptability of the framework to highly heterogeneous GUIs or industrial-scale systems remains unclear. The current experiments mainly focus on relatively constrained domains, which may limit practical significance.

4. Although baseline comparisons are included, the paper does not provide detailed ablation studies or component-level analysis. This makes it difficult to attribute the observed gains to specific design choices.

5. Some parts of the methodology description are high-level, with limited algorithmic or implementation details. This reduces reproducibility and makes it harder for the community to build upon the work.

**Questions:**

1. The framework is framed as original by integrating vision and reasoning. However, many recent works in the last 2–3 years already follow a similar planning–acting–reflecting paradigm. Could the authors clarify how their approach differs in depth and contribution from these existing paradigms?

2. The evaluation covers diverse domains, but some critical scenarios (e.g., large-scale stress testing, real-time responsiveness, highly noisy GUI environments) seem less discussed. Could the authors comment on whether their framework could handle such situations?

3. The experiments are mostly in relatively constrained settings. How confident are the authors that the approach will generalize to highly heterogeneous or industrial-scale GUI environments?

4. The ablation analysis appears limited. Would the authors consider adding further component-level studies to better isolate the contribution of each module?

5. Some technical descriptions remain at a high level. Are there any plans to release more implementation details or supplementary material to support reproducibility?

---

### Official Review · Reviewer_rmpA · 2025-10-31

**Soundness:** 3
**Presentation:** 2
**Contribution:** 2
**Rating:** 4
**Confidence:** 3

**Summary:**

The paper introduces VistaGUI, a multimodal GUI agent that enhances robustness in GUI automation by improving retrieval, perception, and recovery. It integrates three modules: a Parallel Instruction-understanding Module for accurate retrieval, an Adaptive Multi-modal Sensing Framework for dynamic perception, and an Environment-aware State Management system for proactive failure recovery. Experiments on benchmark tasks show that VistaGUI outperforms existing baselines in task success rate.

**Strengths:**

1. The paper is clearly written and well organized, making it easy to follow.
2. The proposed GUI agent systematically addresses three major weaknesses of existing systems—retrieval, perception, and recovery—and demonstrates promising improvements on benchmark datasets.
3. The modular design provides a clear and interpretable framework for GUI automation.

**Weaknesses:**

1. Key components are described at a high level, but critical implementation details are missing. For example, the retrieval procedure is under specified. How are queries formed and results ranked? Similarly, it is unclear how the Reflector detects failures and triggers recovery. Providing these details in the main text or appendix would greatly enhance clarity and reproducibility.

2. The proposed modules mainly extend existing frameworks with additional components. While the system is well-engineered, the conceptual contribution appears incremental, and the improvements stem primarily from architectural integration.

3. Several claims like the proposed method improving recovery speed and overall robustness are not thoroughly validated. Also several additional baselines (e.g., UFO2, GUI-Actor) should be included to provide a more comprehensive evaluation.

4. The paper would benefit from additional qualitative visualizations or case studies to illustrate how the proposed modules work in practice. Moreover, a discussion of potential failure modes would help readers better understand the limitations of the method.

5. Several references are missing or incomplete, especially in Section 4. The paper mentions GPT-4o, OmniParser, PaddleOCR, and baseline methods without proper citations or version details. These should be clearly referenced to ensure clarity and reproducibility.

**Questions:**

1. How exactly is retrieval achieved in the Parallel Instruction-understanding Module?
2. How is failure detection implemented in the Reflector?
3. How does the proposed method perform when used with other backbone models besides GPT-4o (e.g., Claude, Gemini, or open-source models)?

---

### Official Review · Reviewer_Pmj1 · 2025-10-31

**Soundness:** 3
**Presentation:** 2
**Contribution:** 3
**Rating:** 4
**Confidence:** 2

**Summary:**

This paper proposes VistaGUI, which forms a closed loop of retrieval → perception → recovery: it enhances RAG retrieval through parallel intent understanding and stabilizes grounding via multimodal adaptive perception.

**Strengths:**

- Clear closed-loop agent design (retrieval → multimodal perception → rollback/replan) that targets core failure modes.
- The paper introduces a well-founded reflection mechanism, and the proposed PIM further strengthens the pipeline by decomposing multi-intent instructions for parallel retrieval, improving context coverage and downstream reliability.
- competitive gains on multiple Windows benchmarks with ablations.

**Weaknesses:**

I’m not very familiar with this field, maybe from the perspective of researchers working in this area, this might be an interesting paper. Therefore, I’m open to revising my score after considering the opinions of other reviewers.

- It seems that the absolute success rates remain modest; long-horizon and cross-application/generalization (web/mobile) are not fully validated — perhaps such tasks are indeed very challenging?
- In AMF (Adaptive Multimodal Perception), how do we determine which parts of the context are missing, and why not use purely vision-based global perception?
- The paper motivates PIM via the ‘single-objective bias’ of standard RAG/search, and shows SR/Ret.Acc gains. However, it doesn’t quantify how frequently multi-intent instructions occur in WAA/OSWorld-W. Please report the proportion of multi-intent prompts, and give a latency breakdown (per-intent parallel retrieval vs. single-branch). This would sharpen the necessity argument beyond empirical gains.

**Questions:**

see weakness above.

---

### Official Review · Reviewer_verg · 2025-11-01

**Soundness:** 2
**Presentation:** 2
**Contribution:** 3
**Rating:** 4
**Confidence:** 2

**Summary:**

The paper identifies three critical, interconnected failure modes in GUI automation and proposes a corresponding, integrated set of solutions (PIM, AMF, EAM). Most of the experimental results are compelling. VistaGUI achieves a performance lift over strong baselines, and the breakdown by application type (T 2) shows its dominance,'Coding' and 'Windows System'. paper does ablation studies which are important as the pipeline has different choices /components.

**Strengths:**

The paper addresses practical and persistent challenges in GUI-agent reliability through a well-structured modular design. The integration of retrieval (PIM), adaptive perception (AMF), and recovery (EAM) is executed and grounded in clear motivation from real-world agent failures. The experimental setup on WindowsAgentArena and OSWorld-W is thorough, with consistent improvements over baselines, showing that VistaGUI’s modules contribute to more stable and interpretable agent behavior. Overall, the work is timely, and provides useful engineering insights for building good computer use agents.

**Weaknesses:**

Incremental Architectural Advance: While the components seems well engineered with performance, the overall architecture follows a well established modular/hierarchical paradigm (broadly: Planner, Executor, Reflector). The novelty lies more in the specific implementation of the modules rather than a fundamentally new agent architecture. That is models like summarizing interaction history, multi-granularity perception is not new to this method across recent GUI-agent systems such as GUI-Rise, PAL-UI, EARL, and MCPWorld. The new modules (PIM, AMF, EAM) look like solid engineering contributions but I don’t see how they introduce a new reasoning paradigm or learning principle.

The evaluation is conducted exclusively on Windows-based benchmarks if I got it right. It leaves open the question of how well VistaGUI's techniques, particularly the AMF which relies on Windows accessibility APIs, generalize to other platforms like macOS, Linux, or mobile (Android/iOS).

Each component’s effect needs study under controlled conditions (e.g., tool-selection only, summarization only, no recovery). Without this, it’s difficult to attribute what actually drives performance. Please correct me in rebuttal if I missed this in ablation where I found report performance removing one component at a time.

The absence of tests on cross-platform or web-based benchmarks such as Mind2Web, WebArena, or WorkArena limits understanding of how VistaGUI scales beyond its design environment. A discussion on portability or ablation under degraded accessibility signals would strengthen the generalization claim.

**Questions:**

Do you think there can be more discussion on related issues like latency, compute cost, memory overhead, or how the system performs under resource constraints or long-horizon (>100 steps) episodes? This would make it appealing to a broader audience?

While relative improvement is reported, the absolute success rates remain modest (e.g., 32.2% SR in WindowsAgentArena in your summary). A deeper qualitative breakdowns of remaining failures, discussion on practical utility remains unclear as the key contribution seems like creating usable systems.

It’s  bit unclear whether the baseline agents use the same tool access (APIs, DOM access, multi-modal perception modules) as the proposed system. If they dont have of similar access, the performance improvement could stem from tool advantage rather than algorithmic innovation. May be this is mentioned and I missed it.

I am ready to improve score if the concerns above and in weakness are addressed.

---

### Meta-Review · Area_Chair_sU2Y · 2025-12-30

**Summary:**

The reviewers agree that the paper targets a practical problem of robust GUI automation for LLM/VLM-based agents, and that the proposed VistaGUI is a reasonable modular integration of retrieval, adaptive perception, and state management. Several reviewers found the empirical results on WindowsAgentArena / OSWorld-W promising and generally improving over baselines, and appreciated the motivation from real-world GUI-agent failure modes and the interpretability of the modular design.

However, across reviews the main factors keeping the paper below the acceptance threshold are: perceived incremental contribution, reproducibility, evaluation depth, and generalization beyond Windows. Finally, one reviewer (rYRc) raises fundamental concerns about vague/unsupported claims and problem framing shifts, and argues several core ideas appear trivial or insufficiently justified.

**Reviewer Concerns:**

No author rebuttal/response is present.

**Reviewer Scores:**

Given no rebuttal is submitted, reviewers will likely maintain their original negative scores.

Reviewer verg: 4 → 4; Reviewer Pmj1: 4 → 4; Reviewer rmpA: 4 → 4; Reviewer XhzL: 4 → 4; Reviewer rYRc: 2 → 2

---

### Decision · Program_Chairs · 2026-01-26

Reject